# Waste to Carbon: Estimating the Energy Demand for Production of Carbonized Refuse-Derived Fuel

**Paweł Stępień** [1],*, **Małgorzata Serowik** [1], **Jacek A. Koziel** [2] and **Andrzej Białowiec** [1,2]

1   Faculty of Life Sciences and Technology, Institute of Agricultural Engineering, Wroclaw University of Environmental and Life Sciences (Poland), 51-630 Wroclaw, Poland; malgorzata.serowik@upwr.edu.pl (M.S.); andrzej.bialowiec@upwr.edu.pl (A.B.)
2   Department of Agricultural and Biosystems Engineering, Iowa State University (US), Ames, IA 50011-3270, USA; koziel@iastate.edu
*   Correspondence: pawel.stepien@upwr.edu.pl; Tel.: +48-713-205-700

**Abstract:** We have been advancing the concept of carbonized refuse-derived fuel (CRDF) by refuse-derived fuel (RDF) torrefaction as improved recycling to synergistically address the world's energy demand. The RDF is a combustible fraction of municipal solid waste (MSW). Many municipalities recover RDF for co-firing with conventional fuels. Torrefaction can further enhance fuel properties and valorize RDF. Energy demand for torrefaction is one of the key unknowns needed for scaling up CRDF production. To address this need, a pioneering model for optimizing site-specific energy demand for torrefaction of mixed RDF materials was developed. First, thermogravimetric and differential scanning calorimetry analyses were used to establish thermal properties for eight common RDF materials. Then, the model using the %RDF mix, empirical thermal properties, and torrefaction temperature was developed. The model results for individual RDF components fitted well ($R^2 \geq 0.98$) with experimental torrefaction data. Finally, the model was used to find an optimized RDF site-specific mixture with the lowest energy demand. The developed model could be a basis for estimating a net energy potential from the torrefaction of mixed RDF. Improved models could be useful to make plant-specific decisions to optimize RDF production based on the energy demand that depends on highly variable types of MSW and RDF streams.

**Keywords:** waste to carbon; municipal waste; energy recovery; CRDF; torrefaction model; circular economy; refuse-derived fuel; zero waste; waste management

---

## 1. Introduction

Refuse-derived fuel (RDF) refers to the combustible fraction of municipal solid waste (MSW) characterized by high calorific value, which includes common waste groups: plastics, packaging waste, textiles, wood, and rubber [1,2]. RDF is produced in mechanical-biological plants (MBPs) for MSW with dedicated installations to [3,4]:

- recovery of recyclable materials (e.g., plastics, metals, glass, paper),
- biological stabilization of biodegradable waste,
- separation of inert waste, which is destined for the landfill,
- production of fuel (RDF) from high-calorific waste (which cannot be recycled).

RDF is produced from waste, which cannot be reused/recycled yet can be used in energy recovery processes. Production of RDF can minimize landfilling and therefore reduce the potential of methane emissions into the atmosphere [5]. Energy recovery from waste can be enhanced through the initial valorization of the waste stream to increase its calorific value. The process line for the RDF production

consists of several dozen devices usually arranged in series, which perform unit operations aimed at the redistribution of processed waste, enabling its enrichment or removal from the stream (positive or negative separation, respectively) [3]. Current waste management guidelines in the EU promote the recovery of raw material for recycling. As a result, the fuel properties of the RDF will likely deteriorate as high-calorific-value materials (such as plastics, paper, cardboard, textiles) will be sent for recycling.

Thus, a challenge and opportunity exist to valorize a low-calorific value waste in the RDF by increasing its fuel properties via processes such as torrefaction (a.k.a. as "roasting" or "high-temperature drying"). Enhancing energy density of resulting carbonized RDF (CRDF) will increase its attractiveness as an available fuel recovered from waste [6]. This concept fits well into the framework of "waste to carbon", "zero waste", and "circular economy" goals.

Torrefaction is based on the physical and chemical processing of biomass between 200 and 300 °C, mainly at atmospheric pressure under hypoxic conditions. The retention (a.k.a. process) time for typical biomass ranges from 15 to 60 min. Torrefaction reduces the weight (up to ~70%) and volume while maintaining a significant portion of the energy (~90%). Torrefaction practically removes moisture and at least a fraction of volatile organic compounds (VOCs) from the material [7,8]. Torrefaction densifies energy and improves the fuel properties relative to the unprocessed substrate. For example, Białowiec et al. [9] showed that the lower heating value (LHV) of RDF increased from 19.60 to 26.10 MJ·kg$^{-1}$ for CRDF.

The overall working hypothesis is that the purposeful separation and physical/chemical treatment of RDF can be exploited to maximize the energy recovery in CRDF. This is needed for decision-making that considers the evaluation of cost, life cycle analysis (LCA), and for optimizing the CRDF torrefaction that has inherently variable inputs in the available MSW streams. Management of RDF systems deals with high variability of potential CRDF streams, a well-known practical experience that is site-, location-, state-, country-specific. RDF is characterized by high variability in its morphological and quantitative composition. This means that RDF processing via torrefaction could be problematic because the variable RDF mix has different kinetics parameters and energy demand [10]. Thus, the next step in the implementation of the "Waste to Carbon" concept is to create models to estimate the energy demand for torrefaction of inherently variable RDF inputs.

In this research, for the first time, we propose a model for estimating the overall energy demand for the RDF torrefaction. This model includes both the endo- and exothermic processes during torrefaction. We propose to use relatively simple input data that can be obtained by standard methods, i.e., the thermogravimetric analyses (TGA) and differential scanning calorimetry (DSC) [11] for individual materials in RDF mix and their relative percentage weight. The TGA can be used to determine the kinetics of thermal degradation, while DSC can be used for heat flow (both exo- and endo-) measurements during thermal degradation (as a function of programmed torrefaction process temperature). Thus, both the energy released and demand can be measured, and, finally, the overall process energy demand for torrefaction may be estimated. Due to the inherent variability in RDF morphology [12], organic materials may react differently during torrefaction, likely influenced by individual material mix in RDF. Since RDF is a mixture of different organic materials, such as plastics, paper, cardboard, leather, hygienic waste, and biodegradables [13], it is reasonable to hypothesize that different MSW composition can influence the overall energy demand for torrefaction.

This paper presents the formulation of a mathematical model that forecasts the energy demand of the torrefaction. The model considers initial heating (from 20 to 300 °C), and then thermal processing for eight typical types of organic and plastic waste materials in RDF separated during mechanical and biological pre-treatment. The model uses individual material properties. Our working hypothesis that TGA and DSC analyses allow the prediction of the energy demand for individual materials during torrefaction on a macro scale. The proposed approach to modeling can be the basis for assessing the energy consumption of torrefaction of RDF consisted of a given morphological group. The model can be useful in specifying the energy intensity of a given RDF on a macro scale. On its basis, a decision

can be made by MSW plant operators for adjustment of mechanical and biological pre-treatment to the minimization of torrefaction energy demand (Figure 1).

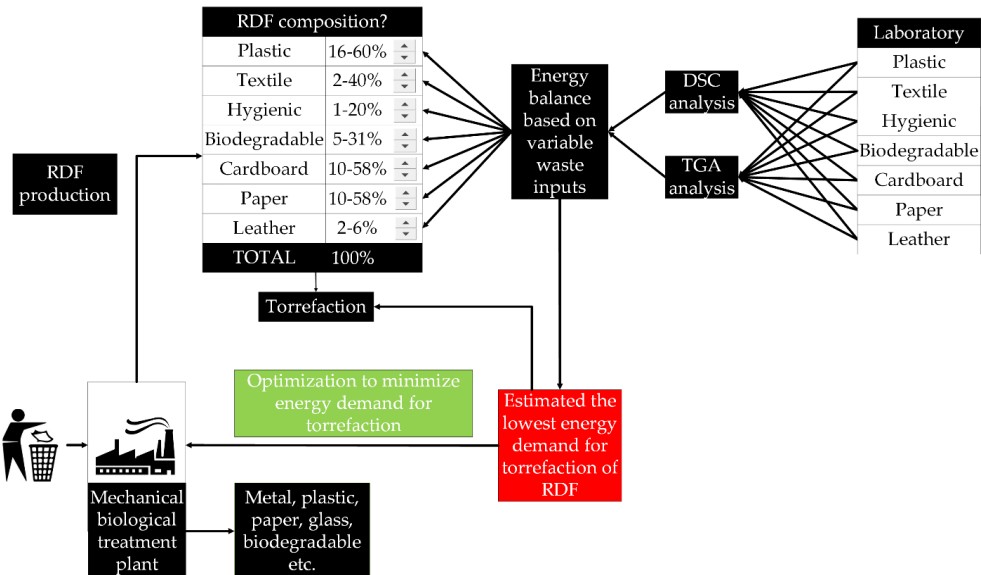

**Figure 1.** The general scheme of the model concept.

## 2. Materials and Methods

### 2.1. Feedstock Preparation

The TGA and DSC were completed on the following RDF materials that are common in MSW: chicken meat (as "biodegradable"), diaper (as "hygienic"), gauze (as "hygienic"), egg packaging (as "cardboard"), paper receipt (as a "paper"), cotton (as a "textile"), genuine leather (as "leather"), polypropylene (PP) (as 2D plastic). All materials were purchased at a large chain supermarket (Wroclaw, Poland), in ~0.5 kg samples of consumer products that terminate as MSW. All samples were dried at 105 °C for 24 h. Then, the waste was ground to ≤0.425 mm size. All samples were dried again just before analysis to release any potential adsorbate from storage.

### 2.2. Thermogravimetric Analysis (TGA)

The TGA was carried out using the stand-mounted tubular furnace previously described by [10]. The change in mass of the sample as a function of time and temperature was measured in triplicates. The temperature increase rate between the initial 20 °C and the final 300 °C was 10 °C·min$^{-1}$, i.e., incorporating a realistic heating phase. Inert $CO_2$ gas was supplied at 10 dm$^3$·h$^{-1}$. The measured percentage of material weight remaining (%$_{TGA, i}$, %) was used for modeling.

### 2.3. Differential Scanning Calorimetry (DSC)

Thermal analysis was performed using the Mettler Toledo DSC 822e differential scanning calorimeter. The endothermic/exothermic changes were tested using the same temperature conditions as for the TGA. Inert $N_2$ was used at 3.6 dm$^3$·h$^{-1}$. The determination was made in 120 μL medium-pressure steel crucibles similar to those described elsewhere [10]. Results were used to develop an experimental database for model inputs of the specific heat ($E_{SH, i, T}$, J·g$^{-1}$).

### 2.4. Modeling Energy Demand of Torrefaction of RDF

The mathematical model was created in the Microsoft Office spreadsheet based on experimental data inputs (available as Supplementary Materials and presented in an organized and accessible format in an accompanying *Data* [14] manuscript). The model outline is shown in Figure 2.

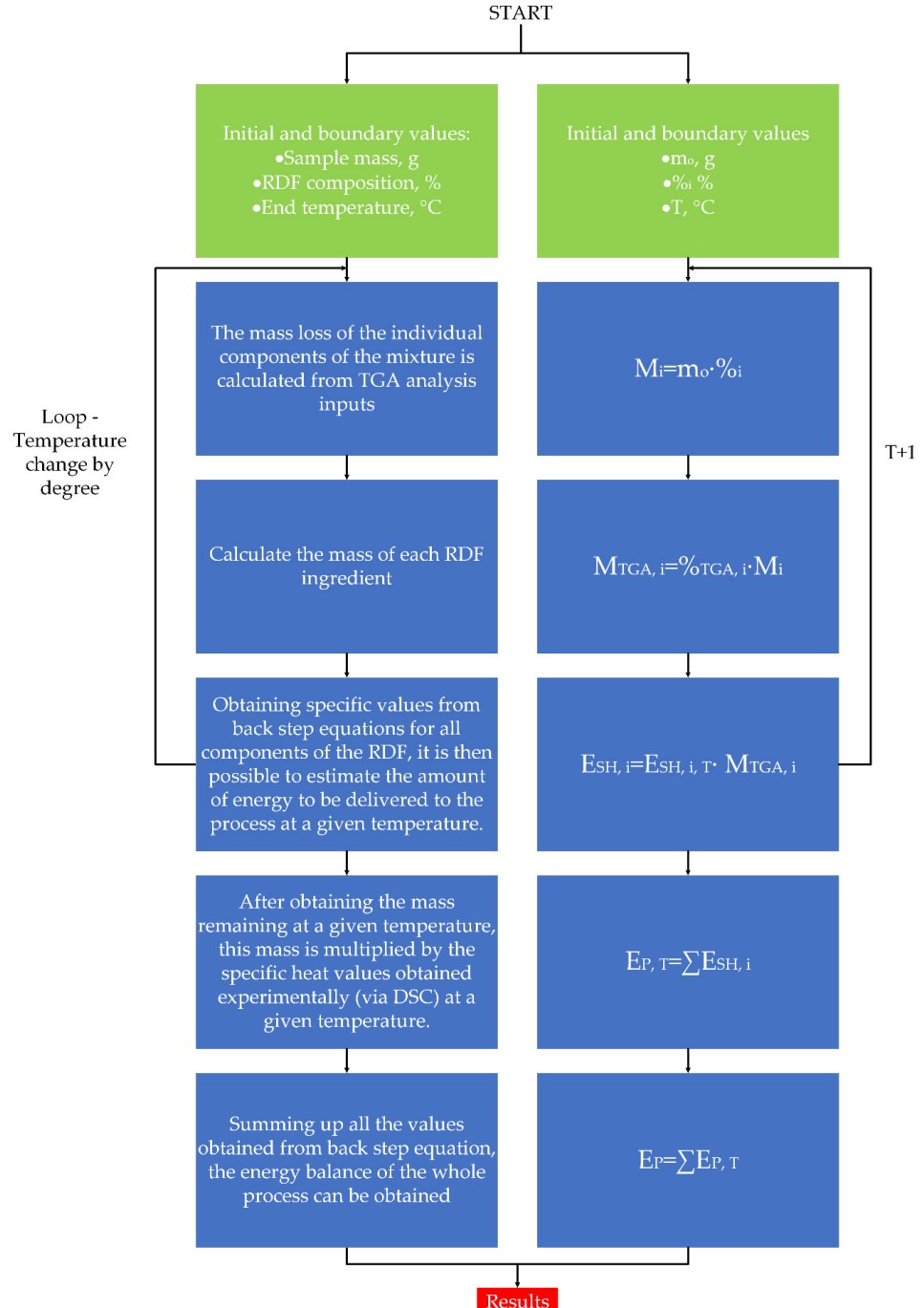

**Figure 2.** The block diagram of the model for estimation of energy demand for RDF torrefaction. Note [14]: RDF = refuse-derived fuel, TGA = thermogravimetric analysis, DSC = differential scanning calorimetry, T = torrefaction temperature, i = site-specific RDF components (e.g., chicken meat, diapers, gauze, egg packaging, paper receipts, cotton, genuine leather, polypropylene), $M_i$ = mass of the RDF component, $m_o$ = the mixed RDF sample mass, $\%_i$ = percentage of individual RDF in a sample, $M_{TGA,I}$ = mass loss of the individual RDF component during torrefaction, $\%_{TGA,i}$ = remaining % of an RDF component after torrefaction, $E_{SH,i}$ = the specific heat of an RDF component at a given process temperature, $E_{SH,i,T}$ = experimental value of the specific heat of an RDF component at a specific temperature, $E_{P,T}$ = process energy in one temperature step, $E_P$ = the energy of the entire process.

The sample weight for model verification was set to 1 g, process temperature from 20 to 300 °C to allow comparison of DSC analyses with the model. Eleven different calculations were carried

out to determine the energy demand of the torrefaction. Ten simulations were performed based on literature concerning the morphological composition of RDF, and one simulation was carried out based on the determined minimum and maximum values of individual waste types groups from the cited publications (Table 1). In the case when any of the RDF material types did not appear in the model, the complement to 100% was calculated according to the existing distribution. The eleventh simulation (optimized RDF) was completed with the "solver tool" built into the spreadsheet using "Generalized Reduced Gradient (GRG) Nonlinear" settings. The exact mathematical formulas used in the spreadsheet solver was based on the model presented elsewhere [15].

**Table 1.** The percentage composition of the RDF mix used for simulations.

| Sample Name | | RDF Composition, % | | | | |
|---|---|---|---|---|---|---|
| | | Plastics | Paper | Diapers | Textiles | Kitchen and Garden Waste |
| RDF 1—UK [16] | | 23.16 | 61.05 | — | 15.79 | — |
| RDF 2—Poland [10] | | 36.17 | 13.45 | 9.86 | 2.67 | 37.74 |
| RDF 3—Canada [17] | | 37.01 | 50.39 | — | 10.08 | 2.52 |
| RDF 4—Poland [18] | | 36.24 | 13.52 | 9.93 | 2.63 | 37.68 |
| RDF 5—Turkey [19] | | 16.90 | 17.10 | — | 66.00 | — |
| RDF 6—Sweden [20] | | 43.00 | 48.00 | — | | 9.00 |
| RDF 7—UK [21] | | 50.55 | 27.18 | — | 12.12 | 10.14 |
| RDF 8—Turkey [12] | | 20.00 | 20.00 | — | 60.00 | — |
| RDF 9—Belgium [22] | | 53.41 | 14.77 | — | 11.36 | 20.45 |
| RDF 10—Germany [23] | | 37.11 | 10.31 | 5.15 | 20.62 | 26.80 |
| Optimized RDF | Min. | 16.90 | 10.00 | 5.00 | 2.20 | 2.46 |
| | Max. | 47.00 | 58.00 | 8.30 | 66.00 | 31.57 |

The solver goal was to find the minimum value of energy needed for torrefaction of the RDF mixture, which the solver selected in the intervals given in boundary conditions for the simulation: temperature from 20 to 300 °C, mass sample 1 g, and RDF composition from Table 1—Optimized RDF. In the case when two or more components were assigned to a given group, the value was divided equally. For example, in RDF 1, the morphological "paper" group (61.05%) has been equally divided into two groups: egg package (30.53%) and a paper receipt (30.53%).

### 2.5. Calculation of the Heating Cost of the Sample

Knowing the energy demand to heat the sample, the cost of heating the sample from 20 to 300 °C has been calculated. For that purpose, the energy demand of the process was divided by the calorific value of the fuel used to heat the sample and then multiplied by the price of the kilogram of the used fuel. For calculations, it was assumed that wood pellets with a calorific value of 17,000 $J \cdot g^{-1}$ and a price of 0.26 $€ \cdot kg^{-1}$ [24], and hard coal with a calorific value of 26,000 $J \cdot g^{-1}$ and a price of 0.22 $€ \cdot kg^{-1}$ were used [25]. The obtained values converted to processing one Mg of each RDFs.

### 2.6. Statistical Analysis

A statistical analysis of variance (ANOVA) was carried out on the differences between the mean values of dependent variables (average remaining weight after torrefaction) on independent variables (a type of RDF material). For the tested variables with normal distribution, the assumption of homogeneity of variance was verified using the Levene test using $p = 0.05$ significance. In addition, a statistical analyses of the linear correlation between experimental and model results were completed.

## 3. Results

### 3.1. RDF Weight Reduction during Torrefaction—TGA

Based on the results of the analysis significant ($p < 0.05$) weight reductions were observed after TGA at 300 °C for an egg package (2.22%) and genuine leather (10.9%) (Figure 3) only. Measured reductions for other materials were statistically ($p < 0.05$) negligible because most organic matter-building compounds started to decompose around or above 300 °C. The lowest reduction was measured for chicken meat, paper receipt and cotton (all at 0.667%).

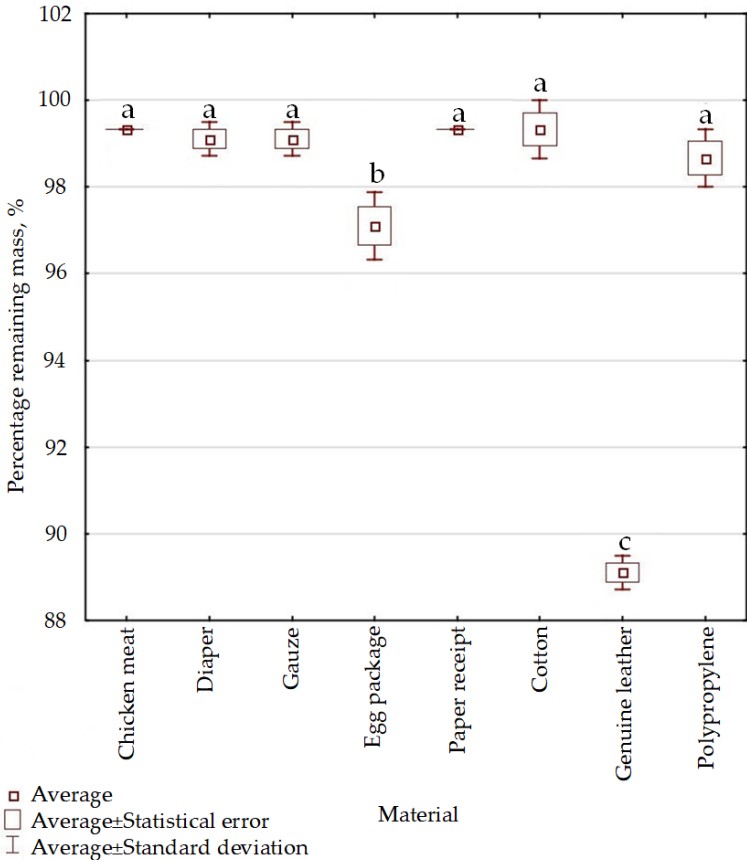

**Figure 3.** The average percentage of the remaining mass of different RDF materials after TGA analysis in the torrefaction temperature range. Letters show significant ($p < 0.05$) differences between each tested material. Letters a, b, and c indicate statistical differences in the obtained results from the whole set, e.g., materials "a" are not statistically different from each other, yet different from both "b" and "c".

### 3.2. Specific Heat during Torrefaction—Evidence of Endo- and Exothermic Reactions—DSC Analyses

Each of the tested materials had a distinct pattern of endo- and exothermic reactions (visualized as "humps" and "valleys" in subsequent figures) during temperature treatment simulating the torrefaction process. This is consistent with the assumption that both types of reactions affect the overall energy demand and that individual RDF materials can be influencing the overall energy demand for the RDF mix. The results are also consistent with the notion that the purposeful separation and mix of RDF materials could be exploited to minimize the energy input into torrefaction. The results below summarize the DSC analyses in the form of specific heat as a function of the torrefaction process temperature for eight common RDF materials. Summary of presented in Table A1 (Appendix A).

Three endothermic reactions were observed during chicken meat analysis, occurring in the following ranges: 171.47–205.52 °C, 245.38–278.31 °C, and 281.29–291.74 °C (Figure 4, Table A1, Appendix A).

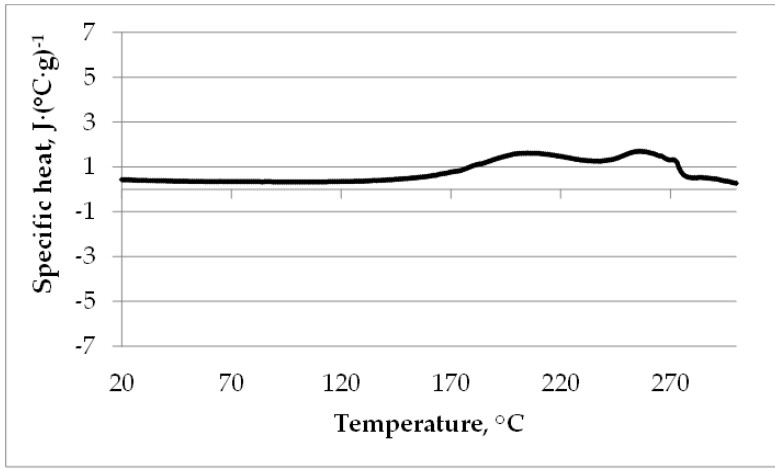

**Figure 4.** DSC characteristics of biodegradable waste (chicken).

For the diaper material, three endothermic changes were observed (Figure 5a, Table A1), occurring in the 124.34–131.35 °C, 156.12–172.11 °C, and from 220 °C (without an apparent completion up to 300 °C) ranges. The gauze material has two reactions. The first endothermic reaction was in the range of 230 °C to 260 °C, followed by a strongly exothermic reaction, which ended at 300 °C (Figure 5b, Table A1).

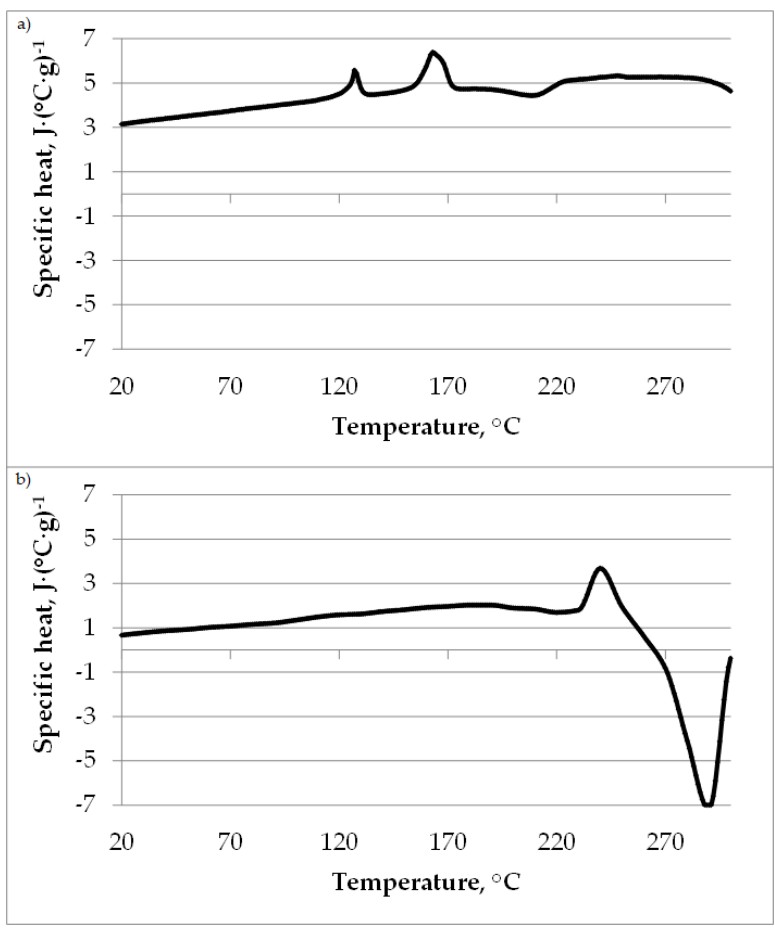

**Figure 5.** DSC characteristics of hygienic waste: (**a**) diaper, (**b**) gauze.

In the case of egg package and paper receipt, two endothermic reactions were observed (in the case of the former, they had a higher value of specific heat). The early reactions were in the 165 °C to 235 °C range, the latter from 235 °C to 290 °C (Figure 6a,b, Table A1).

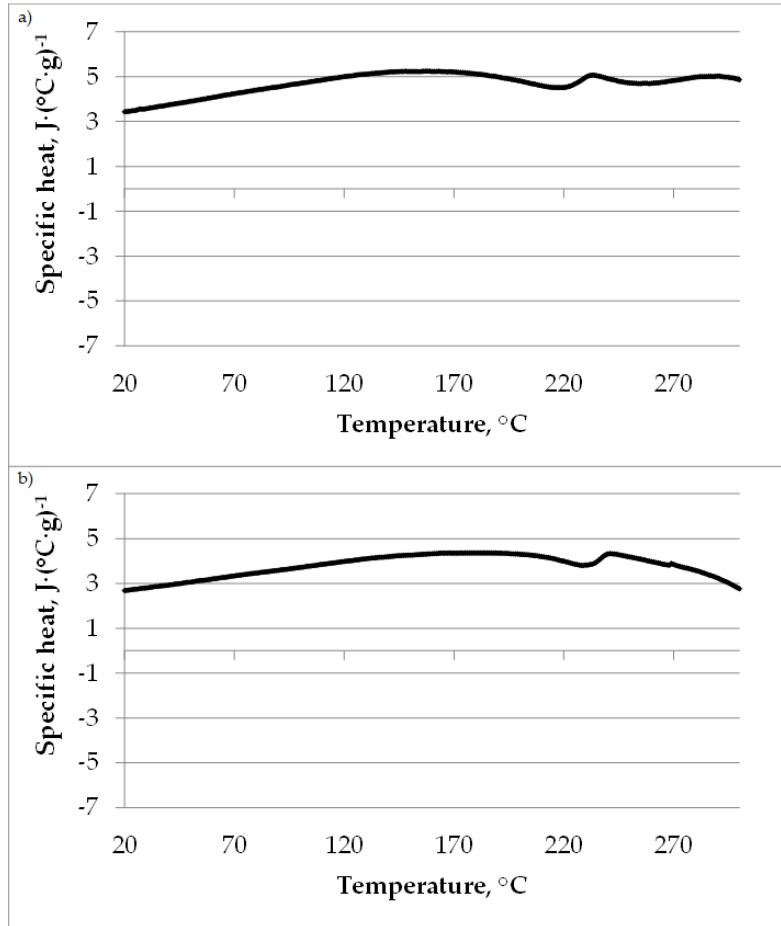

**Figure 6.** DSC characteristics of cardboard waste: (**a**) egg packaging and paper waste (**b**) paper receipt.

During the cotton analysis (Figure 7), two exothermal reactions occurred (Table A1). The earlier one started at 200 °C, and its lowest point was observed at 230 °C when the later reaction began, which ended at 300 °C.

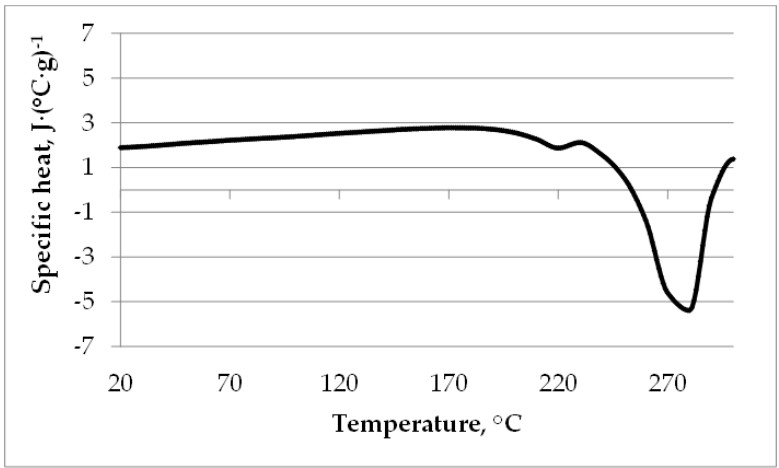

**Figure 7.** DSC characteristics of textile waste (cotton).

Three reactions were observed during genuine leather analysis. They are endothermic, and the latter two are exothermic, which can be observed by the decrease of the instantaneous specific heat value (Figure 8, Table A1).

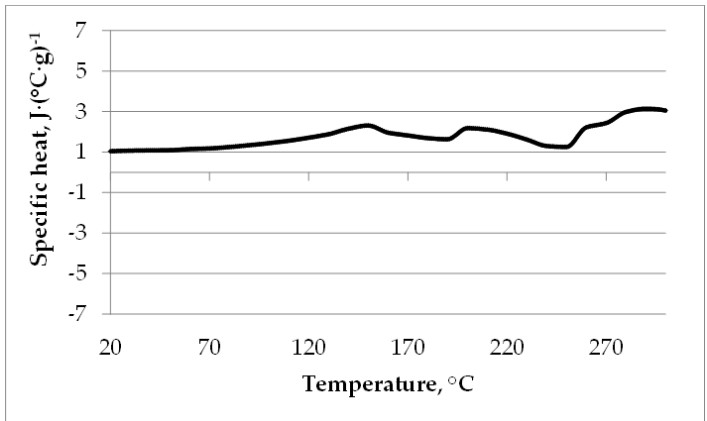

**Figure 8.** DSC characteristics of leather waste (genuine leather).

During the decomposition of PP, one endothermic reaction was observed from 150 to 200 °C (Figure 9, Table A1).

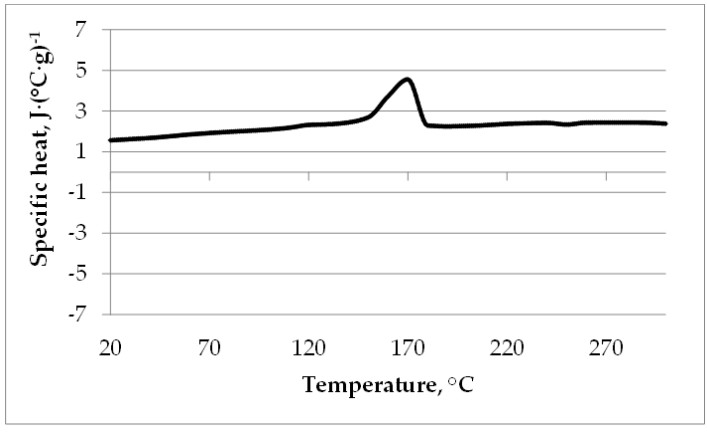

**Figure 9.** DSC characteristics of 2D plastic waste (polypropylene).

### 3.3. Mathematical Modeling Results

The experimental data for the total energy demand for torrefaction of each RDF material type were compared to validate the proposed model (Table 2).

Comparison of individual materials showed that experimental and model results do not significantly differ from each other for chicken meat, paper receipt, and cotton. The highest deviation for total energy demand was observed for genuine leather: 4.34% (20.66 J). The analysis of the linear correlation (coefficient of correlation (R) and coefficient of determination ($R^2$)) between the experimental and model results for chicken meat, gauze, the cotton coefficient were excellent (R = 1.00, $R^2$ = 1.00) followed by the diaper, egg package, paper receipt, genuine leather (R = 0.99, $R^2$ = 0.98).

In the simulation, the solver tool found a solution with the boundary conditions (presented in Table 1) for 10 RDF examples (1–10) and one optimized RDF. The lowest RDF energy demand for torrefaction was 564.05 J·g$^{-1}$ and the optimized RDF consisted of chicken meat (14.28%), diaper (5.00%), gauze (39.42%), egg package (10.00%), paper receipt (10.00%), cotton (2.20%), genuine leather (2.20%), and PP (16.90%). Table 3 summarizes the simulation results for all 10 RDF examples and a resulting, optimized RDF with the lowest energy demand. This illustrates the possibility of exploiting the purposeful separation of RDF with the lowest energy demand.

**Table 2.** Comparison of the total energy demand for torrefaction obtained from DSC analysis (Exp.) and model for individual RDF materials in the treatment range from 20 to 300 °C.

| RDF Material Type | Source of Data | The Total Energy Demand of Torrefaction, $J \cdot g^{-1}$ | % Difference between Experiment and Model |
|---|---|---|---|
| Genuine leather | Exp. | 497.21 | |
| | Model | 476.55 | 4.34% |
| | (Exp.—Model) | (−20.66) | |
| Egg package | Exp. | 1317.18 | |
| | Model | 1312.03 | 0.39% |
| | (Exp.—Model) | (−5.15) | |
| PP | Exp. | 657.68 | |
| | Model | 655.39 | 0.35% |
| | (Exp.—Model) | (−2.29) | |
| Diaper | Exp. | 1278.76 | |
| | Model | 1276.11 | 0.21% |
| | (Exp.—Model) | (−2.26) | |
| Paper receipt | Exp. | 1062.19 | |
| | Model | 1061.29 | 0.03% |
| | (Exp.—Model) | (−0.90) | |
| Chicken meat | Exp. | 215.95 | |
| | Model | 215.90 | 0.03% |
| | (Exp.—Model) | (−0.05) | |
| Cotton | Exp. | 423.86 | |
| | Model | 423.86 | 0.00% |
| | (Exp.—Model) | (0.00) | |
| Gauze | Exp. | 257.45 | |
| | Model | 257.45 | 0.00% |
| | (Exp.—Model) | (0.00) | |

**Table 3.** Specification of the total energy demand of the process obtained from model data for ten types of RDF and one optimized RDF (with the lowest energy demand) in the temperature range from 20 to 300 °C.

| Name Of Sample | RDF Composition, % | | | | | | | | The Total Energy Demand for Torrefaction, $J \cdot g^{-1}$ | The Cost of Heat Production | |
| | Chicken Meat | Diaper | Gauze | Egg Package | Paper Receipt | Cotton | Genuine Leather | PP | | Wood Pellets, €·Mg of RDF | Hard Coal,€·Mg of RDF's |
|---|---|---|---|---|---|---|---|---|---|---|---|
| RDF 1 | 0.00 | 0.00 | 5.26 | 30.53 | 30.53 | 5.26 | 5.26 | 23.16 | 937.27 | 14.3 | 7.93 |
| RDF 2 | 37.74 | 9.86 | 0.89 | 6.78 | 6.78 | 0.89 | 0.89 | 36.17 | 615.57 | 9.41 | 5.21 |
| RDF 3 | 2.52 | 0.00 | 3.36 | 25.19 | 25.19 | 3.36 | 3.36 | 37.02 | 884.75 | 13.53 | 7.49 |
| RDF 4 | 37.68 | 9.93 | 0.88 | 6.76 | 6.76 | 0.88 | 0.88 | 36.23 | 616.21 | 9.42 | 5.21 |
| RDF 5 | 0.00 | 0.00 | 22.00 | 8.55 | 8.55 | 22.00 | 22.00 | 16.90 | 568.43 | 8.69 | 4.81 |
| RDF 6 | 9.00 | 0.00 | 0.00 | 24.00 | 24.00 | 0.00 | 0.00 | 43.00 | 870.84 | 13.32 | 7.37 |
| RDF 7 | 10.14 | 0.00 | 4.04 | 13.59 | 13.59 | 4.04 | 4.04 | 50.56 | 722.51 | 11.05 | 6.11 |
| RDF 8 | 0.00 | 0.00 | 20.00 | 10.00 | 10.00 | 20.00 | 20.00 | 20.00 | 600.00 | 9.18 | 5.08 |
| RDF 9 | 20.45 | 0.00 | 3.79 | 7.39 | 7.39 | 3.79 | 3.79 | 53.40 | 701.13 | 10.72 | 5.93 |
| RDF 10 | 26.80 | 5.15 | 6.87 | 5.15 | 5.15 | 6.87 | 6.87 | 37.14 | 568.57 | 8.69 | 4.81 |
| Optimized RDF | 14.28 | 5.00 | 39.42 | 10.00 | 10.00 | 2.20 | 2.20 | 16.90 | 564.05 | 8.63 | 4.77 |

## 4. Discussion

*4.1. TGA*

For chicken meat, observed reductions were negligible because meat-building proteins decompose at 60 °C to 82 °C [26] and were likely off-gassed during sample drying at 105 °C. Thus, it is recommended that model calibration for the drying effect and/or less destructive methods of drying should be developed in future investigations.

The mass reduction of hygienic waste was negligible. The analyzed hygienic waste diapers and gauze are mainly made of superabsorbent polymers (33%) and cellulose (24%), and lignocellulosic compounds. Research carried out by [27] showed that the thermal decomposition of the lignocellulose compound starts from 200 °C, while the plastics that build absorption polymers start decomposing at ~400 °C [28].

The mass reduction in paper and cardboard group (represented here by the egg packaging), and the paper receipt was also negligible, with the reduction due likely to the degradation of paper pulp, which mainly consists of polymers (cellulose, hemicellulose, and lignin). The degradation of these materials starts from 200 °C to 300 °C. Similarly, as in the case of hygienic waste, a statistically insignificant decrease ($p < 0.05$) in mass was observed as a result of the decomposition of these biopolymers [29].

In the case of cotton, the decomposition process has not practically started. The TGA cotton decomposition characteristics in the inert atmosphere [30] showed that the primary cotton decomposition occurs above 300 °C and is related to the breakdown of α-cellulose bonds.

In the morphological group of textiles, the statistically significant ($p > 0.05$) mass reduction for genuine leather compared with all other materials tested was observed. It amounted to 10.9% weight loss and occurred from 130 to 274 °C, which is ~10% of the sample weight. This finding is consistent with the results obtained by [31] during the breakdown of genuine leather, i.e., the decrease in sample mass was 10% at 300 °C.

The reduction in PP mass was statistically insignificant ($p < 0.05$) and equaled 1.33%. This is likely due because the PP decomposes in 320 to 600 °C range. The observed decrease mass was likely due to the decomposition of polyols, the second of the two main components of polyurethane [32].

*4.2. DSC*

Compared to the DSC results of chicken meat with TGA results, it can be noticed that during the 1st (early) reactions (i.e., occurring earlier as the temperature continued to increase), there was no mass loss. This observation is probably due to the chemical reactions resulting in new products without mass loss [33]. The early conversions may be associated with the reactions of arachidonic acid, whose boiling point is 170 °C. The later conversion could be related to the chemical reaction of erucic acid (b.p. 265 °C). As with arachidonic acid, no reduction in mass was observed during the TGA.

DSC changes in diaper analysis can be related to the melting point of polyethylene plastic (PEHD, 125 °C), which is ~2.2% of the diaper mass and melting of PP (5.8% in diapers), which occurs between 160 °C and 170 °C. Another change is related to cellulose degradation, which is ~24% of the diapers mass [34].

According to the manufacturer (3M, Wroclaw, Poland), the gauze is made of 100% cotton and is bleached with $H_2O_2$. The breakdown of this material is likely related to the decomposition of cellulose fibers and the reactions with the residual $H_2O_2$. The early reactions are likely linked to the distribution of the main component of the gauze, i.e., cellulose and hemicellulose, which decomposes from 220 °C to 400 °C [35].

Two endothermic reactions were observed in the case of egg package and paper receipt. These materials are made of cardboard pulp, the composition of which contains about 86% of lignocellulose, and the observed distribution is characteristic of these compounds [36].

Cotton was similar in composition to the gauze, and the DSC analyses were similar. However, a prominent exothermic reaction was apparent for cotton, likely due to dyes that can affect the thermal decomposition and phase reactions processes [35].

Thermal characteristics of genuine leather, which is made of water, proteins, fats and mineral salts, are more challenging to explain. Since the denaturation of proteins occurs below 100 °C, it cannot be assumed that the peaks are related solely to proteins-related reactions, because of sample drying. It is more likely that they are associated with both the fats breakdown and protein denaturation process. Moreover, agents and dyes added during leather processing can react during torrefaction. The early reactions are endothermic, and the latter two are exothermic, which can be observed by the decrease of the instantaneous specific heat value [27].

During the decomposition of PP, one endothermic reaction was observed, which is associated with the loss of thermal strength (softening, melting), which occurs in the 150 °C to 200 °C range [37].

### 4.3. Mathematical Modeling

The was an excellent correlation between experimental and model results. Small differences between experimental analysis and model results are related to the fact that experimental data of DSC do not consider the TGA results (i.e., mass loss during thermal processing). The mathematical model predicted a mass loss on the base of the TGA analysis. This finding is also reflected in the data presented in Figure 1 (avg. percentage mass loss). The mass loss was the smallest for cotton and gauze, and the largest for genuine leather, where the largest deviation between experimental data was found.

The modeling results of ten different RDF blends (Table 3) showed that the highest energy demand occurs when the mixture consists mainly of paper, i.e., RDF 1 and RDF 3. With more than 60% and 50% share of paper waste, the value of the energy demand for the torrefaction process was 937.27 and 884.75 $J \cdot g^{-1}$, respectively. The results of the torrefaction of raw materials showed high energy demand, e.g., the energy demand for egg package and the paper receipt was 1,061.29 and 1,312.03 $J \cdot g^{-1}$, respectively. Samples with significant chicken meat (RDF 10—26.80%; Optimized RDF—14.28%) and gauze (RDF 5–22%; optimized RDF—39.42%) content were characterized by lower energy demand. RDF 5, RDF 8 should also be considered, in which they did not contain chicken meat, and their energy consumption was 568.43 $J \cdot g^{-1}$ and 600.00 $J \cdot g^{-1}$, respectively. Such a low value is related to the high content of gauze and cotton, which is characterized by low energy demand for processing. The simulations showed that the best materials for the torrefaction process in terms of energy consumption are organic materials of natural origin, which have been processed to a small extent. These include chicken meat, cotton. These materials in their structure contain proteins, fat, and cellulose. Samples with a significant content of the diaper, egg package, paper receipt, and PP increased the energy demand for torrefaction. These materials were made from organic materials. This can be seen in the simulation for RDF 4, where despite 36.68% chicken meat content, the total energy demand was 884.75 $J \cdot g^{-1}$ because of 9.86% was a diaper, 6.76% was egg package, 6.76% paper receipt and 36.23% PP.

A simplified economic analysis carried out showed that with the current prices of wood pellets and hard coal, the application of coal is less expensive (almost twice fold). However, it is also less sustainable. Hard coal has a calorific value (~25 $MJ \cdot kg^{-1}$) that is ~9 $MJ \cdot kg^{-1}$ greater than wood pellets (~17 $MJ \cdot kg^{-1}$). However, considering the ecological aspect, the use of wood pellets has a very low impact on the emission of $CO_2$ into the atmosphere during its combustion. The simulations indicated that for the torrefaction of optimized RDF, the cost of heating of 1 Mg of RDF does not exceed 5 € (Table 3). In the worst-case (RDF—1), the cost of heating during torrefaction is close to 8 €, when coal is used. The presented model allows for initial simulations of costs of RDF torrefaction, which may be useful for the optimization of feedstock preparation for the process.

The developed model is relatively simple and uses RDF material inputs that can be obtained by standard methods. The mathematical model could be further improved and developed through:

(1)  adding a more extensive database of TGA and DSC for other materials present in RDF,

(2)  adding the effects related to the thermal conductivity and scale-up, and

(3)   considering possible interactions during the decomposition of various RDF mixed wastes.

It is also necessary to extend the model by modeling the calorific values of the final torrefaction product (CRDF) and the gas produced in the process, which could be used for energy recovery and adding heat to the process (Figure 10). The final form of the model should also contain the calculations of heat demand for water evaporation. Thus, it would be possible to determine the net energy value of CRDF production due to torrefaction [38].

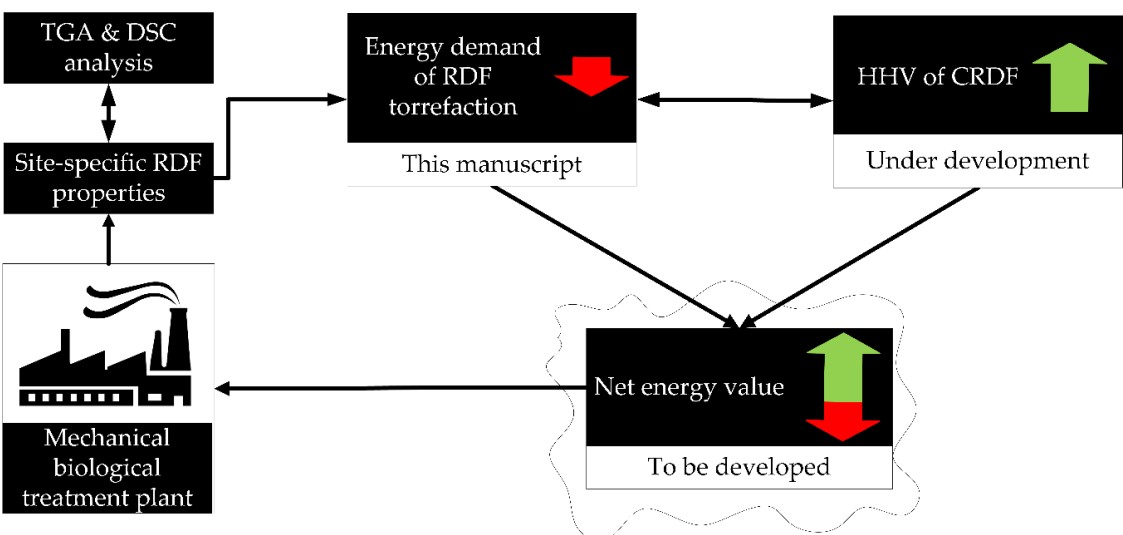

**Figure 10.** The concept for estimating the net energy for torrefaction of RDF and optimization of Waste to Carbon and Waste to Energy for municipal solid waste.

## 5. Conclusions

The energy demand has been identified for eight types of common RDF materials. Relatively small differences of experimentally determined and modeled torrefaction energy demand were found. It was observed that materials containing proteins, fats, and cellulose in their structure are characterized by low energy demand for the process. Therefore, materials such as chicken meat, cotton, gauze should be used to reduce the energy consumption of the torrefaction process. It was observed that if the total percentage of these three substrates is ≥40% energy-consuming, the process drops to about 600 J·g$^{-1}$. Artificially created materials, on the other hand, increase the demand for energy. These include plastics, egg package, diapers, and paper receipts. The combination of TGA and DSC input data from typical types of RDF was a useful tool for determination and modeling of energy demand for torrefaction. This tool allows for an initial analysis of the energy demand of the torrefaction process for a site-specific RDF mixture. In addition, each user can expand the existing database with additional input results (TGA and DSC) if needed. The use of the experimental TGA and DSC inputs in the preliminary determination of the energy demand of the torrefaction process is a reasonable approach because these analyzes can be performed quickly without a need for a specialized laboratory. This, in turn, allows estimation of energy demand and exploit optimization of RDF mix for minimum energy demand. Further research is warranted to improve the model by including a greater number of RDF material types, process scale-up, and heat transfer, and potential interaction between materials in mixtures. Waste to energy and energy recovery from valorized RDF (carbonized RDF) could be estimated to evaluate and optimize a net energy potential from RDF torrefaction for a site-specific application.

**Supplementary Materials:** The following are available online at http://www.mdpi.com/2071-1050/11/20/5685/s1. The developed model (in the form of Excel spreadsheet) is available as Supplementary Material online at [14] (accompanying paper by Stępień et al., 2019).

**Author Contributions:** Conceptualization, A.B., P.S.; methodology, P.S.; formal analysis, A.B., J.A.K.; validation, A.B., J.A.K.; investigation, P.S., M.S.; resources, P.S.; data curation, A.B., P.S., M.S.; writing—original draft preparation, P.S.; writing—review and editing, A.B., J.A.K.; visualization, P.S.; supervision, A.B., J.A.K.

**Funding:** The research was funded by the Polish Ministry of Science and Higher Education (2015–2019), the Diamond Grant program # 0077/DIA/2015/14. "The PROM Program-International scholarship exchange of Ph.D. candidates and academic staff" is co-financed by the European Social Fund under the Knowledge Education Development Operational Program PPI/PRO/2018/1/00004/U/001. Authors would like to thank the Fulbright Foundation for funding the project titled "Research on pollutants emission from carbonized refuse-derived fuel into the environment", completed at the Iowa State University. In addition, this paper preparation was partially supported by the Iowa Agriculture and Home Economics Experiment Station, Ames, Iowa. Project no. IOW05556 (Future Challenges in Animal Production Systems: Seeking Solutions through Focused Facilitation) sponsored by Hatch Act and State of Iowa funds.

**Conflicts of Interest:** The authors declare no conflict of interest.

## Appendix A

**Table A1.** Summary of temperature ranges for endothermic and exothermic processed shown in Figure 4 through Figure 9.

| Transformation No. | 1 | | 2 | | 3 | |
|---|---|---|---|---|---|---|
| **Waste Component** | **Start Point, °C** | **End Point, °C** | **Start Point, °C** | **End Point, °C** | **Start Point, °C** | **End Point, °C** |
| Chicken meat | 171[en] | 205[en] | 245[en] | 278[en] | 281[en] | 291[en] |
| Diaper | 124[en] | 131[en] | 156[en] | 172[en] | 220[en] | > 300[en] |
| Gauze | 230[ex] | 260[ex] | 260[en] | 300[en] | — | — |
| Egg package | 165[en] | 235[en] | 235[en] | 290[en] | — | — |
| Paper | 165[en] | 235[en] | 235[en] | 290[en] | — | — |
| Cotton | 200[ex] | 230[ex] | 240[ex] | 300[ex] | — | — |
| Genuine leather | 120[en] | 145[en] | 150[ex] | 200[ex] | 220[ex] | 260[ex] |
| PP | 150[en] | 200[en] | — | — | — | — |

en—endothermic transformation; ex—exothermic transformation.

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
