# Peer review of "Waste to Carbon: Estimating the Energy Demand for Production of Carbonized Refuse-Derived Fuel"

_sustainability, doi:10.3390/su11205685_

Round 1
Reviewer 1 Report
There are some improvements in the paper, especially in comments on the mixing of individual components of the RDF. The problem is that Authors cannot decide if the paper is on the theory of waste carbonization and optimization of the product (carbonized RDF) or experimental investigation of the carbonization process. I think that physiochemistry of individual waste transformation due to heating to 300 °C is very different for different substances. Discussion of this problem (I am afraid that it is a chemical one) would be interesting. Similarly, I would like to be informed in a clear if carbonization is efficient way of waste processing and which aspects of this efficiency are most important. Unfortunately, I still cannot find the idea which Authors would like to the reader tell.
Author Response
Reviewer 1
There are some improvements in the paper, especially in comments on the mixing of individual components of the RDF. The problem is that Authors cannot decide if the paper is on the theory of waste carbonization and optimization of the product (carbonized RDF) or experimental investigation of the carbonization process.
Authors Response:
Thank you for this comment. The comment was very similar to the overall comments made by other Reviewers. This made us realize that we were not effectively presenting the logic model for this research. Thus, we revised the flow of the Abstract. The revised Abstract addresses also some of the deficiencies identified by other Reviewers and establishes the timeline and process to build and use the proposed model.
Reviewer 1
I think that physiochemistry of individual waste transformation due to heating to 300 °C is very different for different substances. Discussion of this problem (I am afraid that it is a chemical one) would be interesting.
Authors Response:
We agree with the Reviewer. This is indeed what our experimental data shows. These differences for individual RDF components (materials) were measured with relatively simple analyses (TGA and DSC) to be later used as inputs into the model for a mixture of different RDF materials. Of course, the mixture of common RDF materials is a common situation of MSW plant operators. Thus, we built the first model for optimizing site-specific energy demand for torrefaction of mixed RDF materials was developed. First, thermogravimetric and differential scanning calorimetry analyses were used to establish thermal properties for eight common RDF materials. Then, the model using % composition of RDF mix, empirical thermal properties, and torrefaction temperature was developed. Finally, the model was used to find an optimized RDF site-specific mixture with the lowest energy demand.
Our intention was to test (for the first time) applicability of the simplified model based on the mass balance (TGA) and energy balance (DSC), without the need to perform a more sophisticated and detailed chemical transformations analyses. The reason for model simplicity is due to the possible adoption of this approach. We developed the simple model which could be utilized for the decision making by those torrefaction plants operators or decision-makers who need optimize the process from an energy balance and simplified economic analysis (black-box concept), without considering all redundant transformation from that point of view.
Reviewer 1
Similarly, I would like to be informed in a clear if carbonization is efficient way of waste processing and which aspects of this efficiency are most important. Unfortunately, I still cannot find the idea which Authors would like to the reader tell.
Authors Response:
We improved the Abstract and the presentation of the logic behind this research. The answer to this question is highly site-specific, i.e., depending on the type of materials present in a highly-variable stream of RDF. Some municipalities may find that the net economics of CRDF production is favorable due to the high energy recovery and low input costs (also including the energy demand for torrefaction). The developed model could be a basis for estimating a net energy potential from torrefaction of mixed RDF materials. Improved models could be useful to make plant-specific decisions to optimize RDF production based on the energy demand that depends on highly-variable types of MSW and RDF streams. We believe that this message is presented more effectively in the revised manuscript.
Reviewer 1
There are some improvements in the paper, especially in comments on the mixing of individual components of the RDF. The problem is that Authors cannot decide if the paper is on the theory of waste carbonization and optimization of the product (carbonized RDF) or experimental investigation of the carbonization process.
Authors Response:
Thank you for this comment. The comment was very similar to the overall comments made by other Reviewers. This made us realize that we were not effectively presenting the logic model for this research. Thus, we revised the flow of the Abstract. The revised Abstract addresses also some of the deficiencies identified by other Reviewers and establishes the timeline and process to build and use the proposed model.
Reviewer 1
I think that physiochemistry of individual waste transformation due to heating to 300 °C is very different for different substances. Discussion of this problem (I am afraid that it is a chemical one) would be interesting.
Authors Response:
We agree with the Reviewer. This is indeed what our experimental data shows. These differences for individual RDF components (materials) were measured with relatively simple analyses (TGA and DSC) to be later used as inputs into the model for a mixture of different RDF materials. Of course, the mixture of common RDF materials is a common situation of MSW plant operators. Thus, we built the first model for optimizing site-specific energy demand for torrefaction of mixed RDF materials was developed. First, thermogravimetric and differential scanning calorimetry analyses were used to establish thermal properties for eight common RDF materials. Then, the model using % composition of RDF mix, empirical thermal properties, and torrefaction temperature was developed. Finally, the model was used to find an optimized RDF site-specific mixture with the lowest energy demand.
Our intention was to test (for the first time) applicability of the simplified model based on the mass balance (TGA) and energy balance (DSC), without the need to perform a more sophisticated and detailed chemical transformations analyses. The reason for model simplicity is due to the possible adoption of this approach. We developed the simple model which could be utilized for the decision making by those torrefaction plants operators or decision-makers who need optimize the process from an energy balance and simplified economic analysis (black-box concept), without considering all redundant transformation from that point of view.
Reviewer 1
Similarly, I would like to be informed in a clear if carbonization is efficient way of waste processing and which aspects of this efficiency are most important. Unfortunately, I still cannot find the idea which Authors would like to the reader tell.
Authors Response:
We improved the Abstract and the presentation of the logic behind this research. The answer to this question is highly site-specific, i.e., depending on the type of materials present in a highly-variable stream of RDF. Some municipalities may find that the net economics of CRDF production is favorable due to the high energy recovery and low input costs (also including the energy demand for torrefaction). The developed model could be a basis for estimating a net energy potential from torrefaction of mixed RDF materials. Improved models could be useful to make plant-specific decisions to optimize RDF production based on the energy demand that depends on highly-variable types of MSW and RDF streams. We believe that this message is presented more effectively in the revised manuscript.
Reviewer 2 Report
SUSTAINABILITY
Title: Waste to carbon: estimating the energy demand for production of carbonized refuse-derived fuel
Although this is a re-submitted manuscript (ID 587466), my comment will be given will be given in comparison to previous manuscript (ID: 482496).
In the re-submitted manuscript, the authors made the following modifications:
Abstract: line 29-36
Section 2.5., line 153-159
Explanation in section 4.3., line 315-321, 326-333 and 341-342
Conclusions, line 350-355
References No. 24 and 25
According to the Journal instructions – „The abstract should be a total of about 200 words maximum.“ Your abstract contain357 words, you need to shorten the abstract.
Although the authors made changes, some questions remained open.
The discussion section must contain more explanations, it looks like a Results section except section 4.3.
In my opinion, the main drawback of paper is the application of the obtained RDF! You obtained that optimal RDF is with 564.05 J/g. Have you examined which of the torrefactioned RDFs (1-10) give the most energy when using it? Is it optimized RDF (564.05. J/g) the best one?
I think manuscript needs to be supplemented with this experiment.
If the authors disagree or think otherwise, let them explain.
SPECIFIC COMMENTS
Page 5, Write (explain) the meaning of the symbols in the formulas in Figure 2.
Page 13, line 301: “The was an excellent correlation between experimental and model results (both R and R2).” What is the meaning of the R symbol? There are no values of R and R2 in the manuscript.
Page 17, line 473: Delete 29 after numbering 31
Author Response
Reviewer 2
Although this is a re-submitted manuscript (ID 587466), my comment will be given will be given in comparison to previous manuscript (ID: 482496).
In the re-submitted manuscript, the authors made the following modifications:
Abstract: line 29-36
Section 2.5., line 153-159
Explanation in section 4.3., line 315-321, 326-333 and 341-342
Conclusions, line 350-355
References No. 24 and 25
According to the Journal instructions – „The abstract should be a total of about 200 words maximum.“ Your abstract contain357 words, you need to shorten the abstract.
Authors Response:
Thank you for this challenge. We trimmed it down to 200 words while revising the flow of the Abstract. The revised Abstract addresses also some of the deficiencies identified by other Reviewers and establishes the timeline and process to build and use the proposed model.
Reviewer 2
Although the authors made changes, some questions remained open.
The discussion section must contain more explanations, it looks like a Results section except section 4.3.
Authors Response:
Thank you for that comment. Our intention was to test (for the first time) applicability of the simplified model based on the mass balance (TGA) and energy balance (DSC), without the need to perform a more sophisticated and detailed chemical transformations analyses. The reason for model simplicity is due to the possible adoption of this approach. We developed the simple model which could be utilized for the decision making by those torrefaction plants operators or decision-makers who need optimize the process from an energy balance and simplified economic analysis (black-box concept), without considering all redundant transformation from that point of view.
We discussed the TGA and DSC data sufficiently (in our opinion), and we also mentioned that the model is under development and it will be combined with the improved, more comprehensive (holistic) model dealing with the optimization of the calorific value of the final product. The future model should allow evaluating the optimum composition of RDF and torrefaction conditions to obtain the highest value of the ratio of energy yield to energy demand. Therefore at this stage, we decided to propose limited and basic economic analyses conclusions and practical implications. However, we believe that performing more detailed evaluations is required to scale-up of the model. We presented the proposed further steps of our work on the holistic model as a part of the discussion. We believe that we will be able to present more developed and useful (including economic and practical analyses) model.
Reviewer 2
In my opinion, the main drawback of paper is the application of the obtained RDF! You obtained that optimal RDF is with 564.05 J/g. Have you examined which of the torrefactioned RDFs (1-10) give the most energy when using it? Is it optimized RDF (564.05. J/g) the best one?
I think manuscript needs to be supplemented with this experiment.
If the authors disagree or think otherwise, let them explain.
Authors Response:
We revised the Abstract to address this issue, i.e., we showed the logic for model development. First, we had to measure the thermal properties (via TGA and DSC) for torrefied individual RDF components (here 8 common RDF components as a site-specific example). Then, the model using %RDF mix, empirical thermal properties, and torrefaction temperature was developed. The model was tested for individual RDF components first. The model results for individual RDF components fitted well (R2 ≥ 0.98) with experimental torrefaction data. Finally, the model was used to find an optimized RDF site-specific mixture with the lowest energy demand.
We agree with the Reviewer that it would be appropriate to conduct an additional experiment with an optimized mix of RDF components and compared the energy demand with the model. However, we missed this opportunity and cannot conduct it at this time. We are thankful for this insightful comment and will use it as a ‘learning lesson’ for outlining future experiments.
Reviewer 2
SPECIFIC COMMENTS
Page 5, Write (explain) the meaning of the symbols in the formulas in Figure 2.
Authors Response:
We added the definitions of symbols and variables into the caption for Figure 2.
Reviewer 2
Page 13, line 301: “The was an excellent correlation between experimental and model results (both R and R2).” What is the meaning of the R symbol? There are no values of R and R2 in the manuscript.
Authors Response:
Page 11, line 243. R is the ‘coefficient of correlation’ and R2 is the ‘coefficient of determination’, now both definitions added to the text: The analysis of the linear correlation (coefficient of correlation (R) and coefficient of determination (R2)) between the experimental and model results for chicken meat, gauze, the cotton coefficient were excellent (R=1.00, R2=1.00) followed by the diaper, egg package, paper receipt, genuine leather (R=0.99, R2=0.98).
Reviewer 2
Page 17, line 473: Delete 29 after numbering 3
Authors Response:
Thank you for catching this. It is deleted.
Reviewer 3 Report
Overall paper presents an interesting concept, which would be more holistic if connects the experimental, model and practical perspectives altogether. The discussion part might be enriched with a section which is mentioned, but discussed less which is the optimum condition of lowest energy input along with cost minimization in reality.
There are minor areas which might be elaborated for clarification, such as in figure 3, " letters a, b, c indicates statistical difference" requires elaboration on what exactly these letters signify.
A summary of contribution to exothermic and endothermic process by each RDF category would make it more meaningful and understandable to the readers which have been presented from figures 4 through 9 and discussed in text.
Abstract is too long, better to shorten a little. In conclusion, line 353 is a little confusing, needs a little elaboration with simple statement.
Lastly, figure 10 is superfluous in my understanding, mentioning in text is sufficient so that the reader does not have to be under confusion on what is shown here and what is yet to be shown.
Author Response
Reviewer 3
Overall paper presents an interesting concept, which would be more holistic if connects the experimental, model and practical perspectives altogether. The discussion part might be enriched with a section which is mentioned, but discussed less which is the optimum condition of lowest energy input along with cost minimization in reality.
Authors Response:
Thank you for this comment. The comment was very similar to the overall comments made by other Reviewers. This made us realize that we were not effectively presenting the logic model for this research. Thus, we revised the flow of the Abstract. The revised Abstract addresses also some of the deficiencies identified by other Reviewers and establishes the timeline and process to build and use the proposed model.
Our intention was to test (for the first time) applicability of the simplified model based on the mass balance (TGA) and energy balance (DSC), without the need to perform a more sophisticated and detailed chemical transformations analyses. The reason for model simplicity is due to the possible adoption of this approach. We developed the simple model which could be utilized for the decision making by those torrefaction plants operators or decision-makers who need optimize the process from an energy balance and simplified economic analysis (black-box concept), without considering all redundant transformation from that point of view.
We discussed the TGA and DSC data sufficiently (in our opinion), and we also mentioned that the model is under development and it will be combined with the improved, more comprehensive (holistic) model dealing with the optimization of the calorific value of the final product. The future model should allow evaluating the optimum composition of RDF and torrefaction conditions to obtain the highest value of the ratio of energy yield to energy demand. Therefore at this stage, we decided to propose limited and basic economic analyses conclusions and practical implications. However, we believe that performing more detailed evaluations is required to scale-up of the model. We presented the proposed further steps of our work on the holistic model as a part of the discussion. We believe that we will be able to present more developed and useful (including economic and practical analyses) model.
Reviewer 3
There are minor areas which might be elaborated for clarification, such as in figure 3, " letters a, b, c indicates statistical difference" requires elaboration on what exactly these letters signify.
Authors Response:
The use of letters to indicate statistical significance is commonly used in figures. We improved Figure 3 caption to define it:
“Letters a, b and c indicate statistical differences in the obtained results from the whole set, e.g., materials ‘a’ are not statistically different from each other, yet different from both ‘b’ and ‘c’”
Reviewer 3
A summary of contribution to exothermic and endothermic process by each RDF category would make it more meaningful and understandable to the readers which have been presented from figures 4 through 9 and discussed in text.
Authors Response:
We added Table A1 in the Appendix to summarize the data shown in Figures 4 through 9.
Also, such summary exists in the Discussion section “The simulations showed that the best materials for the torrefaction process in terms of energy consumption are organic materials of natural origin, which have been processed to a small extent. These include chicken meat, cotton. These materials in their structure contain proteins, fat, and cellulose. Samples with a significant content of the diaper, egg package, paper receipt, and PP increased the energy demand for torrefaction.”
Reviewer 3
Abstract is too long, better to shorten a little. In conclusion, line 353 is a little confusing, needs a little elaboration with simple statement.
Authors Response:
Thank you for this challenge similarly to comment from Reviewer 2. We trimmed it down to 195 words while revising the flow of the Abstract. The revised Abstract addresses also some of the deficiencies identified by other Reviewers and establishes the timeline and process to build and use the proposed model.
Reviewer 3
Lastly, figure 10 is superfluous in my understanding, mentioning in text is sufficient so that the reader does not have to be under confusion on what is shown here and what is yet to be shown.
Authors Response:
Figure 10 is presented as the last section in Discussion, i.e., a suitable place in the manuscript to forecast where the next possible development could be. We were careful labeling the scope of the current manuscript and the scope of future proposed work. Thus, we respectfully would prefer to keep it in the manuscript. MDPI is generous with space, and graphic representation of ideas can be helpful to non-expert readers.
Round 2
Reviewer 1 Report
I accept Authors response in general, because they are presenting their point of view. I recommend to publish the paper in the present form, even if my perspective is somewhat different.
Author Response
Thank you
Reviewer 2 Report
The authors did their best to improve the manuscript and explained everything clearly in the answer, so it is my decision to accept the manuscript in this form.
Author Response
Thank you
Reviewer 3 Report
In Figure 10, one box states "To be develop" it should be "To be developed"
The manuscript should be thoroughly checked for minor grammatical mistakes.
Author Response
Reviewer: In Figure 10, one box states "To be develop" it should be "To be developed"
Author’s Response: We corrected the wording in Figure 10. Thank you.
Reviewer: The manuscript should be thoroughly checked for minor grammatical mistakes.
Author’s Response: We corrected many minor grammatical issues in the entire manuscript.
This manuscript is a resubmission of an earlier submission. The following is a list of the peer review reports and author responses from that submission.
Round 1
Reviewer 1 Report
The idea presented in the paper is very obvious and should be well known to students of the basic courses of thermochemistry. Additionally, the idea of the heat balance, which is apparently very clear, is presented here in a rather ambiguous way. In my opinion, TG and DSC analysis of conventional materials are very simplified replications of many literature studies. Assuming that properties of “plastic” (?!) – line 100, Figure 1 – may be described by a result of PP determination is intriguing. The paper would be potentially valuable only if equation correlating RDF composition with energy demand for torrefaction would be given, similarly to HHV and LHV which may be found in literature. The next question is why Authors investigated individual materials, instead of typical for waste, the ones mixed with mineral and metallic components ? The result would be the same or different ?
Reviewer 2 Report
There are some improvements that is suggested for this study as following:
1-Line 44: Please also mention that methane has the global warming potential significantly higher than carbon dioxide. The following reference can also be cited for such information "Life cycle assessment of superheated steam drying technology as a novel cow manure management method"
2- Results and discussion sections: Usually the municipal waste has high moisture content and a challenging part of MW management is transportation and storage of the waste. What is the moisture content of the waste you tested? How it can affect the torrefication efficiency? Also how your proposed process helps to address the challenge for the storage and transportation of the waste? Also, one necessary part of the study that is missing in your study is the economic evaluation to show if your suggested process is economically viable.
Reviewer 3 Report
SUSTAINABILITY
Title: Waste to carbon: estimating the energy demand for production of carbonized refuse-derived fuel
GENERAL RECOMENDATIONS
The authors investigate a very interesting topic, briefly, which combination of common RDF waste requires minimal energy consumption for the process of rewriting.
Manuscript is well structured.
The Introduction and Materials and methods are well written.
The abstract must contain more details, especially the best obtained results.
The discussion section must contain more explanations, it looks like a Results section except section 4.3.
Conclusions are written in general, without any results!!!
In my opinion, the main drawback of paper is the application of the obtained SRDF! You obtained that optimal RDF is with 564.05 J/g. Have you examined which of the torrefactioned RDFs (1-10) give the most energy when using it? Is it optimized RDF (564.05. J/g) the best one?
I think manuscript needs to be supplemented with this experiment.
SPECIFIC COMMENTS
Page 2, line 63-64: “For example, the lower heating value (LHV) of RDF increased from 19.60 to 26.10 MJ∙kg -1 for CRDF 63 [9].” If the result from this sentence came from literature 9, please specify Authors, e.g. XY obtained….
Page 4, Write (explain) the meaning of the symbols in the formulas in Figure 2.
Page 5, line 133: DRF change to RDF
Page 5, line 155: “The weight reductions for eight RFD materials were analyzed based on the initial 1.5 g at 20 °C.” – This sentence belongs to Materials and methods section. 1,5 g or 1g?
Page 5-6, line 157-160: These numbers (2.22∙10-3%; 1.09∙10-3% 6.67∙10-3%) is not clear for me when I looking Figure 3.
Figure 3: Meaning of symbols “a, b, c” in Fig. 3?
All text, symbols, numbers in all Figures must have same size font!!!